# Strangely mined bitcoins: Empirical analysis of anomalies in the bitcoin blockchain transaction network

**María Óskarsdóttir**📧⍥*, **Jacky Mallett**📧⍥*

Department of Computer Science, Reykjavik University, Reykjavik, Iceland

⍥ These authors contributed equally to this work.
* mariaoskars@ru.is (MO); jacky@ru.is (JM)

**Citation:** Óskarsdóttir M, Mallett J (2021) Strangely mined bitcoins: Empirical analysis of anomalies in the bitcoin blockchain transaction network. PLoS ONE 16(9): e0258001. https://doi.org/10.1371/journal.pone.0258001

**Data Availability Statement:** The data underlying the results presented in the study are publicly available from Bitcoin.com.

**Funding:** The authors received no specific funding for this work.

## Abstract

The blockchain technology introduced by bitcoin, with its decentralised peer-to-peer network and cryptographic protocols, provides a public and accessible database of bitcoin transactions that have attracted interest from both economics and network science as an example of a complex evolving monetary network. Despite the known cryptographic guarantees present in the blockchain, there exists significant evidence of inconsistencies and suspicious behavior in the chain. In this paper, we examine the prevalence and evolution of two types of anomalies occurring in coinbase transactions in blockchain mining, which we reported on in earlier research. We further develop our techniques for investigating the impact of these anomalies on the blockchain transaction network, by building networks induced by anomalous coinbase transactions at regular intervals and calculating a range of network measures, including degree correlation and assortativity, as well as inequality in terms of wealth and anomaly ratio using the Gini coefficient. We obtain time series of network measures calculated over the full transaction network and three sub-networks. Inspecting trends in these time series allows us to identify a period in time with particularly strange transaction behavior. We then perform a frequency analysis of this time period to reveal several blocks of highly anomalous transactions. Our technique represents a novel way of using network science to detect and investigate cryptographic anomalies.

## Introduction

Blockchain technology contains both structural and operational properties that are designed to safeguard it, including an underlying open decentralized peer-to-peer network between miners, cryptographic protocols, and validation of transactions between users. Its introduction in 2008 has led to a proliferation of cryptocurrencies over the last decade, pioneered by bitcoin [1]. The bitcoin blockchain contains a complete record of over half a billion bitcoin transactions, between over 46 million digital wallets, stored in 670,000 blocks, representing over 18 million bitcoins. The economic impact of this novel technology and the accompanying financial system is already considerable and it has attracted researchers from various disciplines,

**Competing interests:** The authors have declared that no competing interests exist.

including cryptography, economics and network science [2–4], as well as developments into new and diverse applications spaces.

All transactions made using bitcoin are publicly recorded in the blockchain. Owing to this and the dynamic nature of the blockchain, the large number of transactions, numerous wallet and transaction features, and exogenous effects caused by its effective creation of an alternative market based monetary system, it is particularly well suited for network analysis. There are three constructs that can be analysed as bitcoin transaction networks [4]. Firstly, there is the Bitcoin Address Network (BAN), the simplest, where wallets are nodes and transactions make up directed edges. Secondly, the Bitcoin User Network (BUN) unifies wallets that belong to the same user. Finally, the Bitcoin Lightning Network (BLN) is a recently introduced overlay network using a Layer 2 protocol which is attempting to offload transactions from the blockchain itself in order to increase transaction throughput. As the blockchain is growing over time, these networks have become increasingly sparse and peculiar structural properties have emerged [4].

Vallano et al.(2020) summarize research on bitcoin transaction networks, which have been studied to some extent previously [4]. For example, there is an investigation of the acquisition and spending behaviour of bitcoin owners [5]. It has also been observed that the BAN shows evidence of the Pareto principle during the first four years of blockchain, meaning that preferential attachment drove the network's growth and wealth distribution [6]. In updated research, the authors show that preferential attachment still governs the growth of the transaction network, which is now 100 times larger [7]. Two novel contributions perform a data driven analysis of price fluctuations, user behaviour, and wealth accumulation in the bitcoin transaction network, including an investigation of the richest wallets [8] and, an analysis of the transaction network for the first nine years which identified a causal relationships between the movements of bitcoin prices and changes of the transaction network topology [9].

In spite of the blockchain's structural and operational properties which are designed to safeguard it, anomalies, inconsistencies and suspicious behaviour have been observed, and reported. Anomalous behaviour has been connected with colluding miners [10], enhanced performance mining [11, 12], the so-called Patoshi pattern which was detected by Lerner in the first 30,000 blocks [13] and selfish mining, where miners publish the blocks they mine selectively [14]. Another stream of research has focused on detecting anomalies using data driven and machine learning methods, both unsupervised [15–17] and supervised [18, 19]. More recently there has been a stronger focus on network based methods to detect these anomalies, because of the natural structure of transactions [20]. In particular, Elliptic is a cryptocurrency intelligence company focused on safeguarding cryptocurrency ecosystems from criminal activity. introduced a public data set which contains several sub-networks for the blockchain transaction network, with rich node features and labels for licit and illicit transactions. This network has already caught the eye of several researchers [21, 22], who have compared the performance of several supervised learning methods in detecting illicit transactions [23] and address the high class imbalance in the data set using active learning [24].

In this paper we use network science to zoom in on two particular anomalies, which can be seen in the nonce field, in blocks mined in the early years of the blockchain [25]. Given the magnitude of these anomalies—the blocks in question represent well over 3 million mined bitcoin—we investigate whether they may have led to false conclusions about some aspects of bitcoin transactions. More precisely, we develop a methodology to detect cryptographic anomalies and abnormal behavior in bitcoin transactions. It consists of a few steps. Starting with the identification of the anomalous coinbase transactions, we build sub-networks induced by normal and abnormal coinbase transactions. In order to manage the significant scalability and processing issues caused by the size of the blockchain we use sampling strategies. Then we

compute several network measures for the full network and the sub-networks, updating them on a monthly basis. These network measures allow us to compare both the network characteristics, their structural properties and the distribution of some node properties, such as transaction amount and in-degree. Based on this we are able to show that the basic properties of the sampled sub-networks are similar to the full network, making this a feasible approach to analyse big network data. Furthermore, by looking at their evolution over time, we are able to detect periods that need further investigation. Building on our previous work, where the methodology was first presented [26], here in addition to developing it further, we pay special attention to a particularly unusual time period, early in the blockchain which appears to mark the beginning of deliberate dispersal of bitcoin presumably to create the monetary ecosystem. Our results consequently cast some doubt on the origin story of bitcoin, and clearly identify the period in 2010 when bitcoin's use as a monetary unit appears to have been kick started by a large number of transfers originating from coins mined with the cryptographic anomaly we identified.

In the next section we present the methodology we use in this paper, starting with a description of the two anomalies, the sampling techniques developed and network measures, followed by the results in Section, with our results. The paper concludes with a summary of our findings and directions for future work.

## Materials and methods

The methodology of this paper consists of three parts. Firstly, the description of two types of anomalies in coinbase transactions, which is the motivation behind this paper. Secondly, the creation of sub-networks associated with the two anomalies. Finally, the description of network measures which we use to analyse and compare the sub-networks and the full network.

### Background

The now well known origin story of bitcoin is that the technology originated with a posting by a Satoshi Nakamato to the cryptography mailing list in 2008. This was followed by a slow expansion during 2009-10 as early adopters installed mining software and began creating bitcoins, followed by more wide spread adoption following a posting in the slashdot.com online forum in July 2010. Although there has been some question as to whether a single individual could have developed and tested this system, simply due to the range of expertise required, this story has been broadly accepted by researchers.

At the end of 2019 we performed a simple frequency analysis of the hexadecimal values (nibbles) by position, in the bitcoin blockchain. The blockchain itself is an 80 byte block header sequence which is used to both cryptographically certify the transactions belonging to any given mined block, and to provide a proof of work target in the form of a nonce which is used by miners to find a block header that can be used to commit a set of bitcoin transactions. This latter is achieved with a 4 byte nonce, effectively a 32 bit unsigned integer which in the public code is repeatedly incremented by the mining software in order to find a value which results in a double SHA256 operation on the block header that gives a value that is less than the difficulty level governing their mining Difficulty levels are continuously adjusted to maintain a constant rate of mining around 10 minutes/block on average. [25]. Whilst parts of the block header are predictable, notably the version, difficulty and most of the timestamp field, the Merkle-Damgård hash, the previous block hash and the nonce should all be randomly distributed, as they are dependent on properties of the SHA256 algorithm.

Whilst no frequency distribution anomalies were found in either the Merkle-Damgård or the previous block hash, two distinct anomalous patterns were detected in the nonce which is

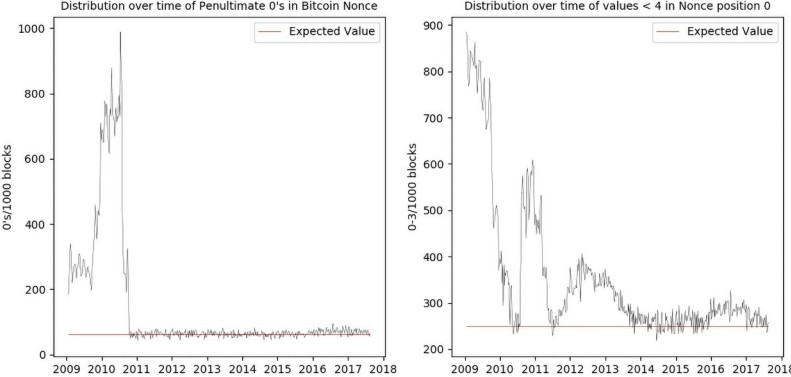

**Fig 1. Anomalous patterns discovered by frequency analysis of the hexadecimal values by position in the bitcoin blockchain.** A: Penultimate Zero Anomaly B: Extended Patoshi Anomaly.

the key component of the proof of work performed by all miners to obtain bitcoins. The bitcoin proof of work performed by miners is simply to repeatedly calculate two SHA256 functions, one of the block header, and the second on the result of the first SHA256. If the numerical result of the second SHA256 operation is less than that specified by the governing difficulty level, then the miner has found a block that can be linked into the blockchain, and receives a specified amount of new bitcoins as a reward.

The two anomalies found with frequency analysis of the individual nibbles of the $2^{32}$ bit nonce field, occur in the first hexadecimal position (nibble) of the block's nonce field as shown in Fig 1B, which in a disproportionate number of blocks has a value in the range 0-3. The second, as shown in Fig 1A is in the penultimate position of the nonce where an abnormal number of 0´s can be seen in the first 18 months of mining. We refer to these as the *P* (extended Patoshi) anomaly and the *Z* (Zerononce) anomaly, respectively.

Both patterns seem to be associated with the originators of bitcoin. The extended patoshi anomaly in the first nibble of the nonce appears in all of the first 64 blocks mined, and is a notable feature of the first months of mining. This was first noticed by Sergio Lerner who observed this feature as part of an analysis on the extra-nonce behaviour in the first year, and attributed this to mining by Nakamato, which seems apparent from its presence in the first blocks mined. Our analysis however also revealed that it returns between mid 2010-11, 2012-14 and 2016-18 as shown in Fig 1B. The second, "penultimate zero", pattern also occurs almost from the beginning of the blockchain, but appears to only occur once, although a very slightly above expected value for zeros in this field is present from 2016.

Although it has been argued online that the patoshi pattern is a consequence of miners evaluating the nonce sequentially, and thus introducing a bias towards lower nonce values, this is not consistent with the expected frequency of valid nonces per block, since in practice these are extremely rare. Courtois et al. (2013) observe that since the nonce value is constrained to 32 bits, the probability of a valid nonce existing for any given block can be expressed as $\frac{2^{32}}{difficulty}$, which would imply on average one valid nonce per block at the easiest difficulty level used until the end of December 2010, and significantly less with the higher difficulty levels used after that. This was verified by an exhaustive search of the nonce space for the first 2000 blocks. [12].

After accounting for the expected number of blocks that would contain these values, (6.25% in the penultimate zero case, and 25% in the patoshi anomaly in the first nibble), we estimate that approximately one third of all coins mined at the first difficulty level are obtained from

blocks mined with these features. Across the entire ten years of both patterns, well over 3 million bitcoins appear to have been obtained from blocks with these distinguishing features.

The size of these two patterns clearly warrants further investigation to see if additional information can be found in the transactions derived from coins mined in these blocks. Previous research into early transactions in the bitcoin network has thrown up evidence of suspicious clusters, notably Shamir and Dorit's work [5] which discovered a large number of coins being progressively consolidated into a small number of apparently connected wallets, however generally research in this area has not had a clear marker of the blocks themselves on which to attach suspicion.

## Induced transaction sub-networks

One of the contributions of this paper is a methodology for extracting specific sub-networks from the blockchain transaction network.

The first step is to prepare the transaction database. For this we extract the entire bitcoin blockchain from origin to November 2019. The data underlying the results presented in the study are publicly available from www.bitcoin.com. We parse the blocks and construct a database of transactions with information about the *from* wallet and one or more *to* wallets. Each transaction corresponds to the movement of bitcoin between wallets. The transactions are furthermore marked with their timestamp and the transaction amount. Wallets that received the miner's reward coins (otherwise known as coinbase transactions) from blocks with the two anomalous patterns are marked as tainted. As coins are transferred to other wallets, the percentage taint for each pattern is calculated and updated for the receiving wallet. The transaction database is thus an edge list of timestamped transactions between wallets, together with the transaction amount and the tainted ratio of both $P$ and $Z$ anomaly. We use the edge list to create a directed network. This type of network is also called the bitcoin address network (BAN) [4]. We focus on the BAN in this research, since we want a representation of the raw transactions between addresses.

The next step in our methodology is extracting specific networks of interest, more specifically, networks that originate with certain coinbase transactions. The process is as follows. We start from the set of all transactions from the origin of the blockchain, until a given time point and use this data to create a BAN. From this BAN we consider sub-networks induced by specific coinbase transactions. This entails snowball sampling where we start from a set of coinbase transactions, follow their edges to the linked wallets, which are added to the sub-network together with the transactions. Subsequently, any wallet in the full network that is linked via a transaction to one of the most recently added wallet in the sub-network, is also added to the sub-network. This process is repeated until no more transactions can be added. Since the full network is static and directional, the process will terminate.

Due to the size of the entire blockchain it is not feasible to build the sub-networks with the snowball sampling technique using all the specific coinbase transactions under consideration. To mitigate this, we choose a random sample from the considered coinbase transaction to start the snowball sampling with. To get more robust results this is repeated several times.

In this paper, we apply our proposed methodology to the two anomalies that were identified in the coinbase transactions, namely the $Z$ and the $P$ anomaly, and compare their induced sub-networks to the full network and the sub-network that does not stem from either of the two anomalies. We thus consider three sets of coinbase transactions to induce our sub-networks as listed below.

$^T\mathcal{Z} = \{\text{cb}|\text{ The } Z \text{ anomaly is in the nonce of the cb block}\}$

$^T\mathcal{P} = \{\text{cb}|\text{The } P \text{ anomaly is in the nonce of the cb block}\}$

$\neg\mathcal{Z} \cap \neg\mathcal{P} =$ {cb|Neither the *Z* anomaly nor the *P* anomaly is in the nonce of the cb block}

As a result, we obtain, in addition to the full network –which we refer to as *All*– three sets of sub-networks, each one induced by the sub-sets of transactions listed above. We refer to these as *Tainted Z*, *Tainted P* and *Not Tainted Z & Not Tainted P*, respectively. We build these sub-networks and the full network incrementally, first using transactions from the origin until January 2010 and then in each iteration adding one more month until May 2012. When inducing each sub-network, we randomly sample 1000 of the respective coinbase transactions and repeat the process ten times. In the Results section, we show the mean of these ten repetitions. When we take a closer look at the last months of 2010, we build the networks at more frequent intervals, with 1-4 days between increments.

## Network measures

The objective of this paper is to compare the structure and properties of the full BAN to the sub-networks induced by tainted and non tainted coinbase transactions. Below, we describe the network measures which we include in our analyses.

First we measure basic properties of the networks. The three fundamental measures are the number of nodes, density and diameter [27]. *Number of nodes* is simply the total number of nodes in the network. *Network density* is defined as the number of edges divided by the maximum possible number of edges. It gives an indication of how well connected the network is. Finally, *network diameter* is a measure of the length of the longest shortest path in the network. Given a pair of connected nodes in a network, there is a path between them that is shorter than any other path between them. The diameter is the longest of such paths in the network. Similar to the diameter of a circle, it gives the longest distance to connect any two nodes. In our analyses we calculated the network diameter based on a random sample of 1000 pairs of nodes, because of the time complexity when finding the shortest path between all pairs of nodes.

In their study of transaction dynamics in the BAN, Kondor et al. (2014) used the Gini coefficient to quantify inequality in the network [6]. Generally, the Gini coefficient is defined as

$$G = \frac{2\sum_{i=1}^{n} ix_i}{n\sum_{i=1}^{n} x_i} - \frac{n+1}{n} \tag{1}$$

where $\{x_i\}$ is a monotonically non-decreasing ordered sample of size *n*. Thus, $G = 0$ indicates perfect equality, or every observation being equal in terms of the value being considered, whereas $G = 1$ indicates complete inequality. In this paper we use the Gini coefficient to characterize the heterogeneity of the distribution of in-degree, out-degree, tainted *Z* ratio, tainted *P* ratio and transaction amount of the nodes in the full network and sub-networks.

Kondor et al. (2014) also investigated structural properties of the network in terms of assortativity and clustering coefficient [6]. Assortativity or degree correlation of the network measures the nodes' tendency to be linked to nodes with a similar degree [27]. It is obtained using the Pearson correlation coefficient of the out- and in-degrees of connected node pairs

$$r = \frac{\sum_e (j_e^{out} - \overline{j^{out}})(k_e^{in} - \overline{k^{in}})}{L\sigma_{out}\sigma_{in}} \tag{2}$$

where for the edge *e* that links node $v_{from}$ to $v_{to}$, $j_e^{out}$ is the out-degree of node $v_{from}$ and $k_e^{in}$ is the in-degree of node $v_{to}$,

$$\overline{k^{in}} = \sum_e k_e^{in}/L \quad \text{and} \quad \sigma_{in}^2 = \sum_e (k_e^{in} - \overline{k^{in}})^2/L. \tag{3}$$

$\sigma_{out}$ and $\overline{k^{out}}$ are computed in a similar way. An assortative network (where $r > 0$) is characterized by high degree nodes being linked to other high degree nodes and low degree nodes being linked to other low degree nodes. In contrast, in a disassortative network ($r < 0$) high degree nodes have a tendency to connect to low degree nodes, creating a hub and spoke structure.

The clustering coefficient of a network is defined as the density of triangles in the network, or

$$C = \frac{1}{N} \sum_v \frac{2\Delta_v}{d_v(d_v - 1)} \tag{4}$$

where $\Delta_v$ is the number of triangles with node $v$ and $d_v$ is the degree of node $v$. The sum runs over all nodes in the network [27]. To compute $C$ we must ignore the directionality of the network. The clustering coefficient measures how connected the nodes are in their closest neighborhoods.

These measures are computed for each full and sub-network as they are incrementally built from month-to-month. As a result we obtain times series showing the development of the networks' properties.

## Results

### Trends in the early years of blockchain

We start by looking at the properties of the sub-networks in comparison to the All network. Fig 2 shows the diameter, number of nodes and density for the networks as subsequent months are added. Note the log scale on the y-axis. Firstly, and not surprisingly, the All network has the most nodes, however as we consider a longer timespan, the sizes of the sub-networks grow in the same manner as the All network. Secondly, the density of the sub-networks is higher than that of the All network. This is expected because of the way the sub-networks are

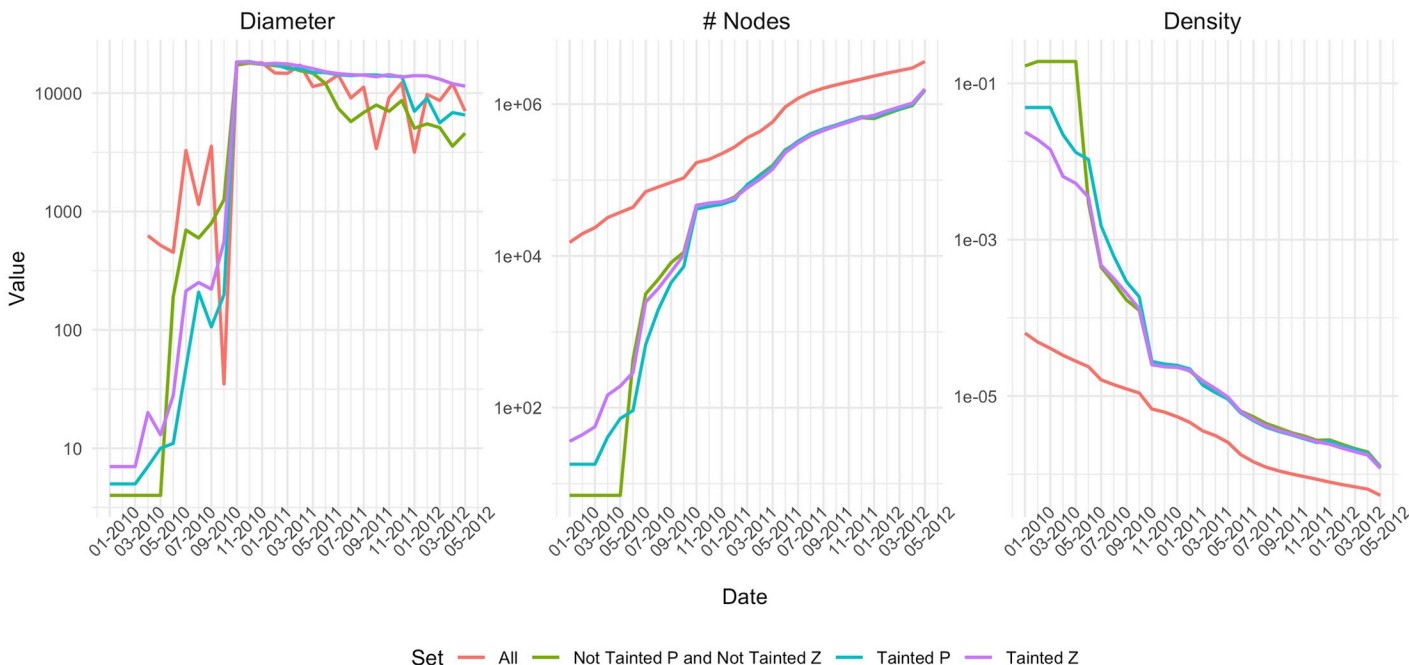

**Fig 2. Evolution of diameter, number of nodes and density in the network of all transactions and in the three sub-networks.**

constructed. At the beginning, each source node induces an almost fully connected network, but as more nodes are added, the number of edges is proportionally lower, and thus the density decreases. Finally, the diameter is rather fuzzy in the beginning, but as the networks grow in size, the diameter becomes similar for all of them. This indicates that the sub-networks span a similar range as the All network. To conclude, our proposed way of constructing sub-networks induced by a sample of coinbase transactions, seems to generate networks that are comparable to the All network.

Next we look at the structural properties of the All and sub-networks, including the distribution of equality. Figs 3 and 4 show the Gini coefficient for in-degree, out-degree, transaction amount, tainted $Z$ and tainted $P$, on the one hand, and the degree correlation and clustering coefficient, on the other hand, for the All network and each of the three sub-networks as months are added incrementally. In each plot, the red line denotes the whole network. We can see how the values for the sub-network all converge towards to each other and are slowly nearing the red line. The distance between them can probably be attributed to the way the sub-networks are created. Moreover, we see that in the beginning, the in-degree tends to be more equally distributed in the sub-networks than in the whole network, whereas there is an opposite behavior for out-degree, the distribution of out-degree is less equal in the sub-networks. Kondor et al. (2014) speculated that the reason for the Gini being high for in-degree and low for the out-degree, was that at the beginning of the blockchain, people were collecting their coins into one wallet, since they were unable to exchange them [6]. In our case, the reason for the Gini being low for the in-degree and high for the out-degree can be explained by the way the sub-networks are created. When adding a transaction to the sub-network, its prior transactions are not added, so it is expected that the in-degree for all newly added transactions are similar, since new nodes start from 'square zero'. We note that the Gini of the out-degree converges to the full network ahead of the others, implying that the behavior of the first few months is due to the building of the sub-network.

Next we look at the Gini coefficient of the Tainted $Z$ and Tainted $P$ ratio. For all sub-networks, the Tainted $Z$ Gini remains higher than in the All network, and they converge early on. This implies that these coinbase transactions get distributed in the transaction network quickly. The Tainted $P$ Gini is higher in the sub-networks at first, but in October 2010, the All network takes over. The Gini of the Tainted $Z$ is higher than that of the Tainted $P$ in the sub-networks and the full network. Regarding the inequality in terms of amount, we see that at the beginning both Tainted $P$ and Tainted $Z$ sub-network have very high values, indicating a very unequal distribution of wealth in these sub-networks. However, the Gini value quickly drops and then remains below the Gini of the full network.

We can see from Fig 4 that in 2010 all the networks have a rather high clustering coefficient, which decreases as time goes on. The clustering coefficient is comparable in the All and the sub-networks. The degree correlation fluctuates a lot during the time period we consider, especially in the sub-networks. There it also remains higher than in the full network until early 2011. Both sub-networks of not tainted transactions have a high clustering coefficient in the beginning, whereas all converge to the same low value towards the end of the period. This indicates that the structural properties of the networks we consider vary greatly between themselves and also across time, which gives cause for further investigation.

The development of the distribution of inequality in the sub-networks compared to the full network shows how the tainted coinbase transactions blended in with the rest of the transactions in the blockchain. Our analysis helps identify peculiarities in the transaction network at certain moments in time where the transaction network ought to be investigated in more detail. For example, the development of the networks' degree correlations raises questions, because of the varied patterns in the sub-networks. In addition, there is a considerable change

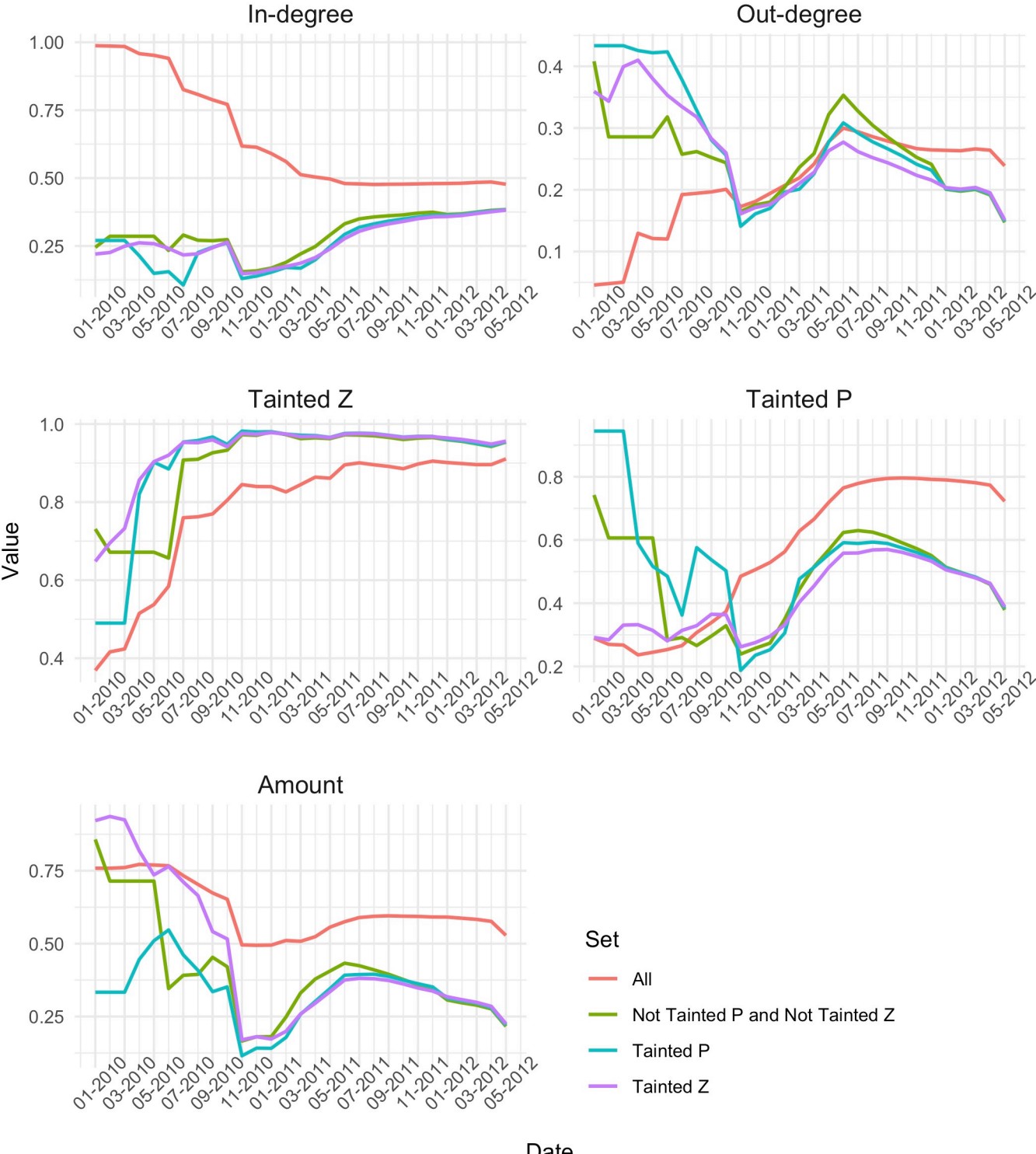

**Fig 3. Evolution of Gini coefficients of in-degree, out-degree, tainted *Z* ratio, tainted *P* ratio and transaction amount, for the all transaction network and three sub-networks.**

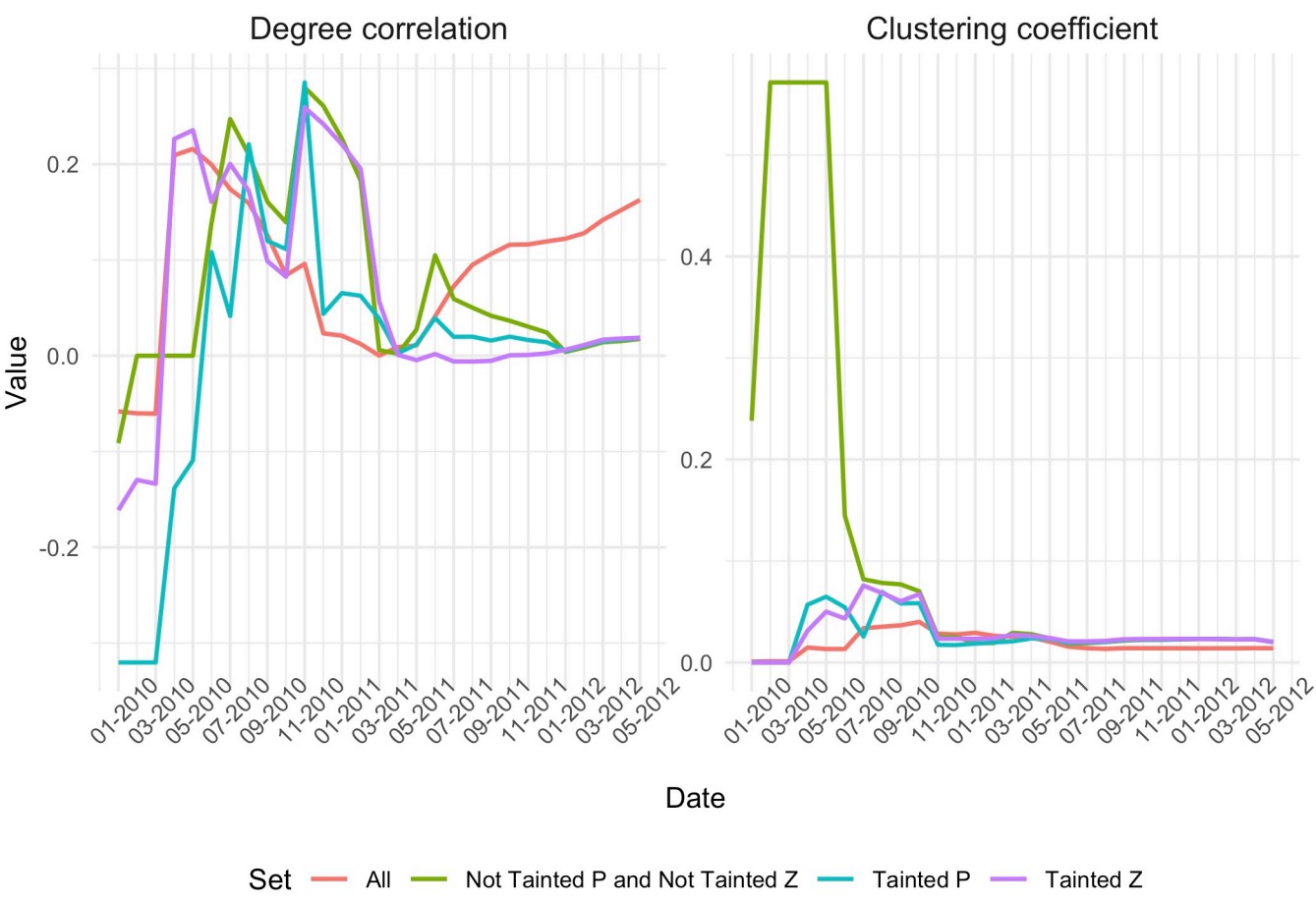

**Fig 4. Evolution of degree correlation and clustering coefficient for the all transaction network and three sub-networks.**

in all the measures around November 2010. The tainted *Z* ratio seems to be least affected by this, however. We will take a closer look at this behavior in the next subsection.

### A closer look at November 2010

In our analysis so far, we witnessed a shift in both the Gini measures and the network structural measures in the final quarter of 2010. Therefore we will take a closer look at the months October, November and December of 2010. We repeat our analysis from before, this time with smaller time steps and more granularity. Fig 5 shows the Gini values at a more granular level and Fig 6 shows the same for the degree correlation and the clustering coefficient of the full network and the three sub-networks, for the months October, November and December 2010. These values are obtained by increments of 1-5 day in each step.

We see from these figures that the shift happens around November 15th and that it is a rather drastic shift. For example, in Fig 5, the in-degree Gini coefficient of the full network changes from close to 0.8 til almost 0.6. For the full network, the Gini decreases in terms of in-degree, out-degree and amount, but increases in terms of tainted *Z* and tainted *P*. The sub-networks show a similar trend, except for tainted *P* where their values decrease after the middle of November, in contrast to the full network. The change is more drastic in the sub-networks than in the full network when looking at out-degree, tainted *P* and amount. It is interesting to look at the development of the tainted *P* inequality in the tainted *P* network. Before the shift, it

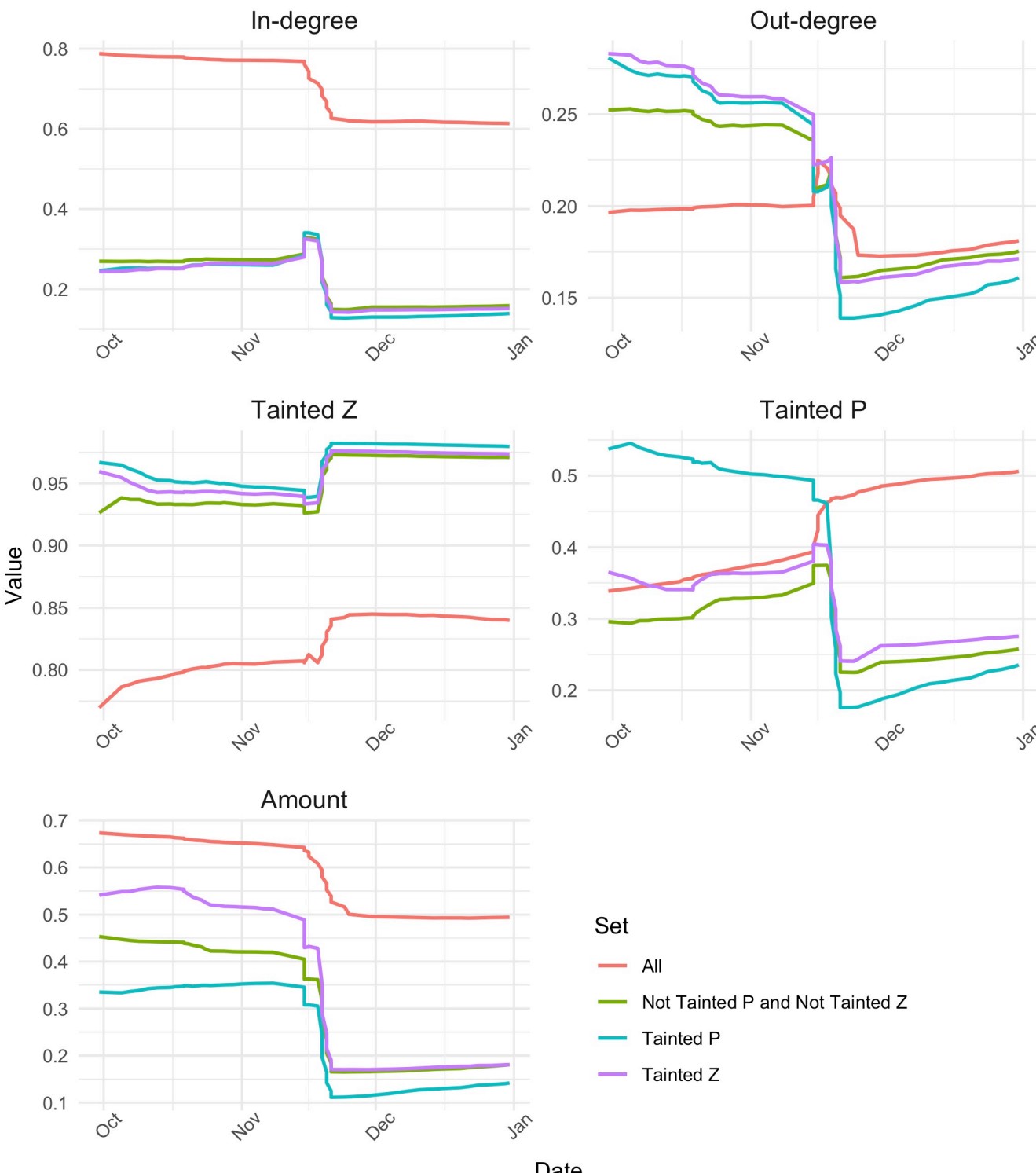

**Fig 5. Evolution of Gini coefficients of in-degree, out-degree, tainted *Z* ratio and tainted *P* ratio and transaction amount in the full transaction network and three sub-networks in October, November and December 2010.**

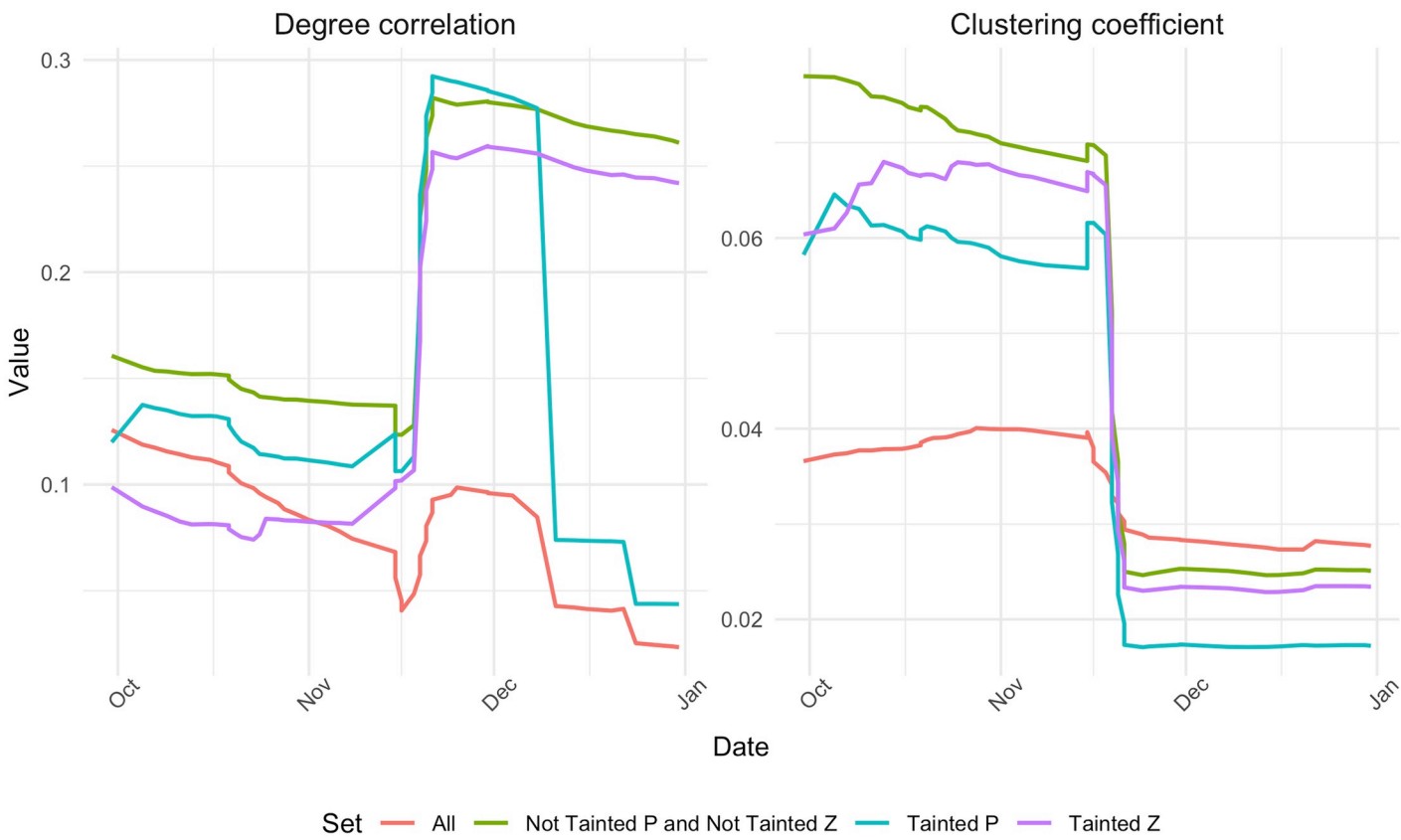

**Fig 6. Evolution of degree correlation and clustering coefficient in the full transaction network and three sub-networks in October, November and December 2010.**

is very high, above 0.5, but it takes a large dive around mid November and is the lowest in all networks. At the same time, the tainted *P* inequality increases overall, i.e. in the full network.

In terms of the structural measures, see Fig 6, the clustering coefficient drops in all networks, and relatively more in the sub-networks than in the full network. This implies that many transactions are being added, which dilutes the ratio of triangles and thus the clustering is reduced. We also see here that the degree correlation fluctuates more than the other measures. The tainted *Z* and not tainted sub-networks are similar in their trends, with a big increase. However, both the full network and the tainted *P* sub-network, take a sudden dip on November 15th, then they increase (the increase is bigger in the sub-network) before going down again in the first half of December. This similarity in behavior, again indicates that the *P* anomaly needs closer inspection.

**Transaction count analysis.** Following this analysis we sampled blocks mined during this period and their associated transactions manually. Another way to examine the evolution of the use of bitcoin as a monetary unit is to simply look at the number of transactions associated with each block. The creation of bitcoin blocks is independent of the number of transactions, the blockchain difficulty level is automatically adjusted to cause bitcoin blocks to be created on average every 10-12 minutes. This, in conjunction with the requirement that all miners must see all transactions that will be committed by the winning block, is what determines the upper limit on the total number of transactions that any block can contain. In later years this is 3-4000 transactions/block. In the first year of mining however the majority of blocks only had a

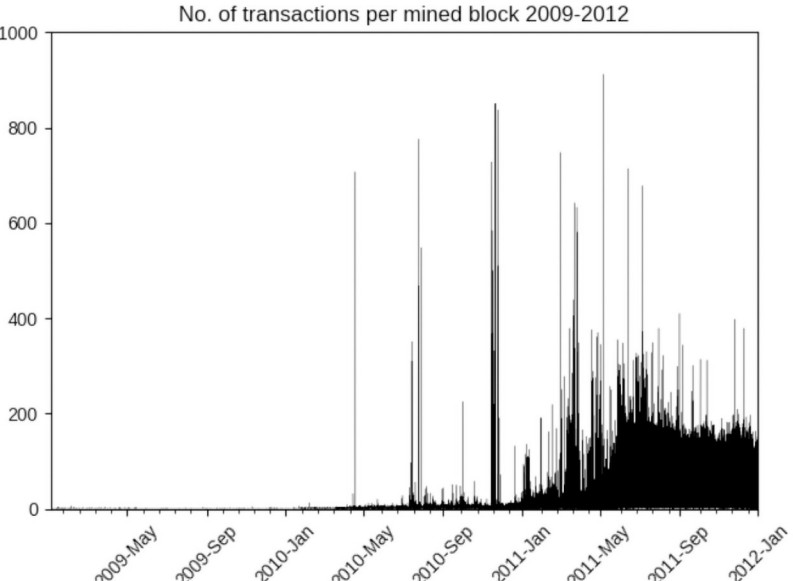

**Fig 7. Average number of transactions/block mined between 2009-2011.**

single transaction, the coinbase transaction awarding the miner of that block with the mined bitcoins, as very few transactions between bitcoin holders were performed. This pattern continued into early 2010 as shown in Fig 7.

Fig 8 focuses on the period in the second half of 2010 identified by the preceding network analysis. Rather than a gradual increase in transactions over time, as might have been expected if bitcoin adoption followed a diffusion process as interest spread among enthusiasts, we see isolated instances of very large numbers of transactions being made extremely quickly, often

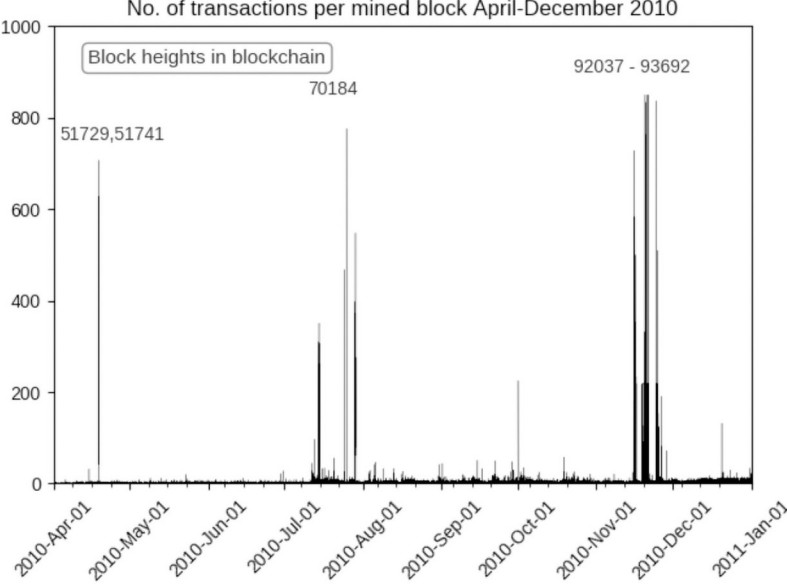

**Fig 8. Average number of transactions/block mined in period of interest in 2010.**

committed in the same or consecutive blocks, which implies they were made within the same ~12 minutes. Following each of these instances there is a marked increase in the average number of transactions until November 2010 when starting on November 15th at 18:45:30 (block height 92037) there is a two week period of bursts of blocks with large numbers of transactions, corresponding exactly with the time period identified by the above network analysis.

All of these large bursts of transactions are heavily sourced from tainted coins from both patterns, and manual examination shows interesting and distinguishable characteristics with the transactions in these blocks, notably large numbers of transfers of the same amount, transfers going immediately through a wallet which is never used again, and in the early blocks, notably 51728 and 51729 a series of transfers each precisely 0.01 bitcoins less then the previous one, although originating from different wallets. The earlier and smaller bursts may indicate testing of the software that was presumably used to create these transactions, it seems highly improbable that these were performed manually given the short time frame, and number of transactions made. For example, block 51729 https://www.blockchain.com/btc/block/ 000000001786abd75dc912d8eabe85080c7e822858d445644fa3a3e059c2033b. This activity appears to begin early in 2010, with 6 transactions made on block 35637, shown in Fig 9. There then appear to be three distinct instances of these disbursements in 2010, what appears to be a short burst on 1st April 2010, a larger instance in July following which average transaction activity begins to noticeably increase, culminating with a major set of transactions in November 2010, beginning on the 15th the same period identified by the network analysis as marking a noticeable shift in the Gini coefficient and other measures.

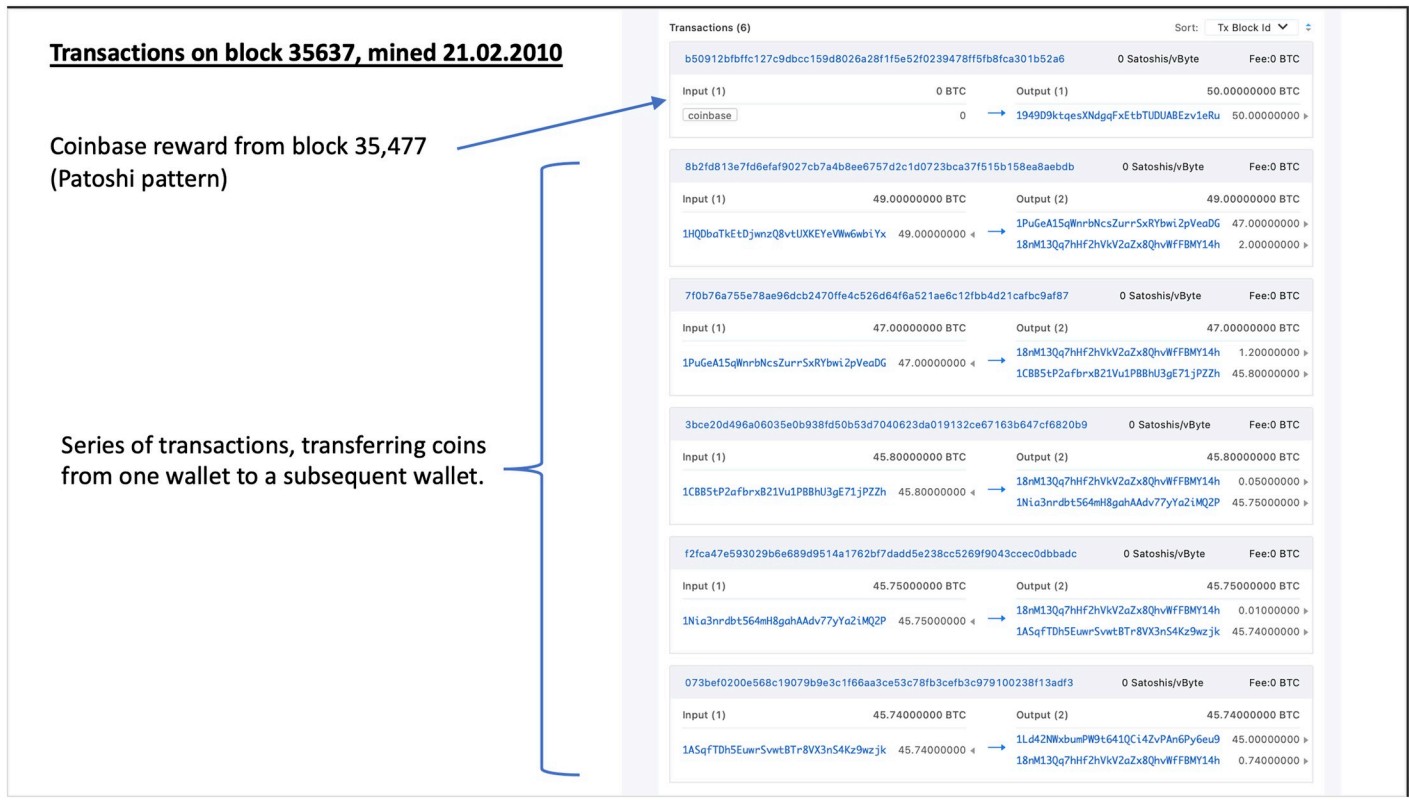

**Fig 9. Transactions on block 35637, mined 21.02.2010.**

## Conclusion

Analysis of the entire transaction network for any cryptocurrency is prohibitively expensive both in CPU and disk time, frustrating what would otherwise be an ideal target for network science. If this form of monetary unit is to be adopted widely then its integrity must be verifiable. Finding an anomaly in the cryptographic underpinnings is not particularly useful in itself, without being able to investigate how coins related to that anomaly subsequently behaved.

In this paper we used network science to look at the evolution of several network measures and distribution of transaction properties in the bitcoin transaction network to investigate the prominence of two anomalies which stem from coinbase transactions. We presented a methodology for constructing sub-networks induced by certain bitcoin transactions using sampling which allowed us to adequately estimate the networks' properties. We compared the networks' structural characteristics to the full network and saw that the distribution of several node properties, such as in-degree, transaction amount and tainted ratio is different in the sub-networks when compared to the full network. This is apparent in the networks until late 2010, when they start to converge to what is observed in the full network. In particular, degree correlation of the sub-network with both anomalies shows a great deviation from the rest at the same time as both these anomalies were prominent in block mining. Based on this information we then examined transactions in the period we had identified more closely, and also performed a simple frequency analysis which clearly illustrated the highly anomalous transaction behaviour around the dates identified by the network analysis.

The size of the blockchain and its transactions places a prohibitively high computational complexity on analysing its network behaviour, hence using this approach as a basis for similar methods to compress computation time for block chain transaction analysis is worth exploring. In contrast to anomaly detection methods which aim at detecting specific anomalous transactions, our technique is meant to investigate the entire transaction network with the goal of finding abnormal behavior in its structure, as measured by various network measures. This approach can help narrow down the set of transactions that need to be investigated further as we did in this paper, since it is difficult to label each and every transaction as anomalous or not.

Further work is needed to get a better understanding of the networks we examined and the bitcoin transaction network. We saw in our analyses that the more frequent updates of the development of network measures gave more detailed insights, and we could see better when and how the anomalies are having an effect on transaction patterns. We would like to carry out our analyses for the entire blockchain at this more granular level. Also, we have only analysed transactions until mid 2012. In our continued work, our plan is to consider the entire blockchain, and investigate the recurrence of the P anomaly in 2012-13 and 2016-17. Finally, we included only a handful of network measures in our analyses. Many other exist, which could be included in a follow up study.

## Acknowledgments

The authors thank Arnthór Logi Arnarson and Alexander Snær Stefánsson for their contribution which made this project possible.

## Author Contributions

**Conceptualization:** María Óskarsdóttir, Jacky Mallett.

**Data curation:** Jacky Mallett.

**Formal analysis:** María Óskarsdóttir, Jacky Mallett.

**Methodology:** María Óskarsdóttir, Jacky Mallett.

**Visualization:** María Óskarsdóttir, Jacky Mallett.

**Writing – original draft:** María Óskarsdóttir, Jacky Mallett.

**Writing – review & editing:** María Óskarsdóttir, Jacky Mallett.

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
