## [Decision Letter · Decision Letter 0]

9 Jun 2021

PONE-D-21-11382

Strangely Mined Bitcoins: Empirical analysis of anomalies in the Bitcoin Blockchain Transaction Network

PLOS ONE

Dear Dr. Óskarsdóttir,

Thank you for submitting your manuscript to PLOS ONE. After careful consideration, we feel that it has merit but does not fully meet PLOS ONE’s publication criteria as it currently stands. Therefore, we invite you to submit a revised version of the manuscript that addresses the points raised during the review process.

We look forward to receiving your revised manuscript.

Kind regards,

Hocine Cherifi

Academic Editor

PLOS ONE

Journal Requirements:

2) Please include a link to the source of the Bitcoin dataset in your methods section.

3) Please include captions for your Supporting Information files at the end of your manuscript, and update any in-text citations to match accordingly. Please see our Supporting Information guidelines for more information: http://journals.plos.org/plosone/s/supporting-information.

4) Thank you for submitting the above manuscript to PLOS ONE. During our internal evaluation of the manuscript, we found significant text overlap between your submission and the following previously published works, some of which you are an author.

- https://link.springer.com/chapter/10.1007%2F978-3-030-65351-4_46

Please revise the manuscript to rephrase the duplicated text, cite your sources, and provide details as to how the current manuscript advances on previous work. Please note that further consideration is dependent on the submission of a manuscript that addresses these concerns about the overlap in text with published work.

Reviewers' comments:

Reviewer's Responses to Questions

**Comments to the Author**

1. Is the manuscript technically sound, and do the data support the conclusions?

Reviewer #1: Yes

Reviewer #2: Partly

2. Has the statistical analysis been performed appropriately and rigorously? 

Reviewer #1: Yes

Reviewer #2: Yes

3. Have the authors made all data underlying the findings in their manuscript fully available?

Reviewer #1: Yes

Reviewer #2: Yes

4. Is the manuscript presented in an intelligible fashion and written in standard English?

Reviewer #1: Yes

Reviewer #2: Yes

5. Review Comments to the Author

Reviewer #1: Review commnets on PONE-D-21-11382:

"Strangely Mined Bitcoins: Empirical analysis of anomalies in the Bitcoin Blockchain Transaction Network"

The authors studied the coinbase transaction network to find abnormal behavior in its structure, as measured by the heterogeneity of various network measures: in-degree, out-degree, tainted Z ratio, tainted P ratio, transaction amount, assortativity, and clustering coefficient.

This research topic is fascinating and important, and the paper is well written. In addition, the structure of the study is appropriate. The method of finding general anomalies at coarse time intervals and then examining the details at fine time intervals is very interesting. However, some explanations are somewhat difficult to understand, as I comment below. In addition, the interpretation of the analysis results seems to be a little weak in the current manuscript.

(1) comment 1

In l.92-l.97 (page 3), the authors wrote that "One anomaly occurs in the first hexadecimal position (nibble) of the block's nonce field as shown in Fig.1 B which is a disproportionate number of blocks has a value in the range 0-3, and the other shown in Fig. 1A is in the penultimate position of the nonce where an abnormal number of 0's can be seen in the first 18 months of mining. We refer to these as the P (Patoshi) anomaly and the Z (Zerononce) anomaly, respectively".

However, the reviewer does not fully understand why these are anomalies. Please elaborate on the reason these are recognized as anomalies.

(2) comment 2

In l.133-l.134 (page 4), the authors wrote that "We start from the set of all transaction from the origin of the blockchain, until a given time point and use this data to create a BAN."

Please elaborate on why the authors concentrate on analyzing Bitcoin Address Network (BAN) instead of Bitcoin User Network (BUN). The reviewer understands that BUN is more important than BAN because each user owns many addresses to manage their crypto asset.

(3) comment 3

In l.208 (page 6), the author wrote, "These measures are computed for each full and sub-network as they are incrementally built from month-to-month."

The author also wrote in l.230 (page 6) that "for the All network and each of the three sub-networks as months are added incrementally."

If the reviewer's understanding is correct, the authors add transaction data on a monthly basis, and the network grows larger and larger. However, Bitcoin transactions are temporary and often do not involve long-term relationships, as in, for example, purchase and sales transactions between companies.

Therefore, the reviewer is concerned that networks constructed to include past transactions may introduce undesirable bias in the results of anomaly detection. The reviewer asks the author to explain that this question is not raised in the paper.

(4) comment 4

In l.332-l.338 (page 9), the author wrote "These may indicate software tests, it seems rather improbable that these were manual transactions.3 There appear to be three distinct instances of these disbursements in 2010, what appears to be a short test on 1st April 2010, a larger instance in July following which average transaction activity begins to noticeably increase, culminating with a major set of transactions in November 2010, beginning on the 15th the same period identified by the network analysis as marking a noticeable shift in the Gini coefficient and other measures".

The reviewer wants the authors to clearly explain what kind of software testing is discovered by P anomaly and Z anomaly. Do other sources report any significant changes in the Bitcoin system during this period? Or have there been any reports of any illegal activities?

For this reason, the reviewer thinks that the interpretation of the analysis results seems to be a little weak in the current manuscript.

Reviewer #2: The authors analyze the bitcoin transaction network via complex network tools, which in my opinion is interesting because it allows detecting topological properties of the system otherwise hidden. However, I have two major concerns about the work.

The first one is about the abstract. I think it is not very clear, a reader should approximately know from it the major findings and the approach used, but the authors simply state to find generic "anomalies" which is clearly not enough. I think that the abstract must be totally rewritten because the current form is not a guideline for the main paper.

The second concern is more methodological. The authors' work is heavily based on the Gini coefficient; however, it is known that the Gini coefficient has a bias dependent on the sample size. In this system, the sample size varies over time, I doubt that such temporal variation might have a big component that comes from the sample size bias. I would suggest the author integrate some other heterogeneity measure and show if they observe a consistency, if not they should go into details and explain why. Although it is not the first use of the Gini coefficient I believe that a reliability check is still necessary.

6. PLOS authors have the option to publish the peer review history of their article (what does this mean?). If published, this will include your full peer review and any attached files.

Reviewer #1: No

Reviewer #2: No

---

## [Author Response · Author response to Decision Letter 0]

9 Aug 2021

Reviewer #1: 

1.The authors studied the coinbase transaction network to find abnormal behavior in its structure, as measured by the heterogeneity of various network measures: in-degree, out-degree, tainted Z ratio, tainted P ratio, transaction amount, assortativity, and clustering coefficient.

This research topic is fascinating and important, and the paper is well written. In addition, the structure of the study is appropriate. The method of finding general anomalies at coarse time intervals and then examining the details at fine time intervals is very interesting. However, some explanations are somewhat difficult to understand, as I comment below. In addition, the interpretation of the analysis results seems to be a little weak in the current manuscript.

 Authors’ response: Thank you for reviewing our paper. We agree that the research topic is both fascinating and important, and are pleased to hear that you found it interesting.

2.comment 1

In l.92-l.97 (page 3), the authors wrote that "One anomaly occurs in the first hexadecimal position (nibble) of the block's nonce field as shown in Fig.1 B which is a disproportionate number of blocks has a value in the range 0-3, and the other shown in Fig. 1A is in the penultimate position of the nonce where an abnormal number of 0's can be seen in the first 18 months of mining. We refer to these as the P (Patoshi) anomaly and the Z (Zerononce) anomaly, respectively".

However, the reviewer does not fully understand why these are anomalies. Please elaborate on the reason these are recognized as anomalies.

 Authors’ response: We thank the reviewer for pointing this out, and apologize for not including this content, it was clearly needed. A clarification as to why this is anomalous has been added. Essentially, we would expect values of the nonce to be randomly distributed across the blockchain, and during the periods highlighted in this paper, they are not.

3.comment 2

In l.133-l.134 (page 4), the authors wrote that "We start from the set of all transaction from the origin of the blockchain, until a given time point and use this data to create a BAN."

Please elaborate on why the authors concentrate on analyzing Bitcoin Address Network (BAN) instead of Bitcoin User Network (BUN). The reviewer understands that BUN is more important than BAN because each user owns many addresses to manage their crypto asset.

 Authors’ response: We thank the reviewer for this very valid question and apologize for not being clear enough about this aspect in our manuscript. For our research purposes, we are not interested in the actual users, but simply the transactions and the transaction patterns. Therefore, we focus on the BAN as it represents the raw transactions between wallets. It would be really interesting in a follow up study to examine the BUN, which requires implementing heuristics in order to cluster addresses, and would thus change the structure of the network.

4.comment 3

In l.208 (page 6), the author wrote, "These measures are computed for each full and sub-network as they are incrementally built from month-to-month."

The author also wrote in l.230 (page 6) that "for the All network and each of the three sub-networks as months are added incrementally."

If the reviewer's understanding is correct, the authors add transaction data on a monthly basis, and the network grows larger and larger. However, Bitcoin transactions are temporary and often do not involve long-term relationships, as in, for example, purchase and sales transactions between companies.

Therefore, the reviewer is concerned that networks constructed to include past transactions may introduce undesirable bias in the results of anomaly detection. The reviewer asks the author to explain that this question is not raised in the paper.

 Authors’ response: We are grateful to the reviewer for pointing out this lack of clarity in our manuscript. The anomaly we are inspecting is a property of the money that is being transacted, but the transaction itself is not anomalous. The anomaly is generated when the bitcoins are mined and can be viewed more as seeds that diffuse taintedness through the network. As the bitcoins are bought and sold, the anomaly is diluted and we calculate the percentage taint for each transaction. This percentage is the property that we are measuring using the Gini coefficient in Figs. 3 and 5 in the Tainted p and Tainted z fields. 

When calculating the various network measures, we are doing it to investigate the structural properties of the entire transaction network, in comparison to the transaction sub-networks that originate with either tainted coinbase transactions or regular coinbase transactions. To inspect this evolution, we should indeed include the entire history of transactions, to see how the networks evolve. As such, our approach is fundamentally different from regular anomaly detection. 

5.comment 4

In l.332-l.338 (page 9), the author wrote "These may indicate software tests, it seems rather improbable that these were manual transactions.3 There appear to be three distinct instances of these disbursements in 2010, what appears to be a short test on 1st April 2010, a larger instance in July following which average transaction activity begins to noticeably increase, culminating with a major set of transactions in November 2010, beginning on the 15th the same period identified by the network analysis as marking a noticeable shift in the Gini coefficient and other measures".

The reviewer wants the authors to clearly explain what kind of software testing is discovered by P anomaly and Z anomaly. Do other sources report any significant changes in the Bitcoin system during this period? Or have there been any reports of any illegal activities?

For this reason, the reviewer thinks that the interpretation of the analysis results seems to be a little weak in the current manuscript.

 Authors’ response: We have added an example of some of these transactions as a figure (Fig. 9), as well as more detail on the actual blocks, which can be explored by interested readers using online viewers, and clarified the text around the suggestion that the earliest smaller bursts are software testing. More examples can certainly be provided, however the next set of transactions features wallet addresses whose owners have been publicly identified, and we feel it might be inappropriate to publish these in this context.

Other sources, notably the Shamir paper we mention, have reported anomalous series of transactions, which we believe overlap with those we have found, but without the benefit of the markers that we are able to attach thus facilitating the network analysis. Other papers in economics have also detected structural changes in macro characteristics in the network in this period, but again without being able to make the association with anomalous mining.

As far as legality is concerned, we think this goes outside of the scope of this paper, although the topic could certainly make an interesting work in and of itself. As far as we are aware, it is not illegal to use a private mining technique on the blockchain, and publish an alternate one. 

Reviewer #2: 

The authors analyze the bitcoin transaction network via complex network tools, which in my opinion is interesting because it allows detecting topological properties of the system otherwise hidden. However, I have two major concerns about the work.

 Authors’ response: We are grateful for your review of our paper which has helped us improve it.

1.The first one is about the abstract. I think it is not very clear, a reader should approximately know from it the major findings and the approach used, but the authors simply state to find generic "anomalies" which is clearly not enough. I think that the abstract must be totally rewritten because the current form is not a guideline for the main paper.

 Authors’ response: We thank you for this valuable comment. We have rewritten the abstract and hope that it is now clearer.

2.The second concern is more methodological. The authors' work is heavily based on the Gini coefficient; however, it is known that the Gini coefficient has a bias dependent on the sample size. In this system, the sample size varies over time, I doubt that such temporal variation might have a big component that comes from the sample size bias. I would suggest the author integrate some other heterogeneity measure and show if they observe a consistency, if not they should go into details and explain why. Although it is not the first use of the Gini coefficient I believe that a reliability check is still necessary.

 Authors’ response: We thank you for drawing our attention to this important drawback of the Gini coefficient. To the best of our knowledge, this known bias of the Gini coefficient occurs for small sample sizes. As the middle figure in Fig. 2 in our manuscript shows, this is not the case in our analyses, except possibly for the first three months where the sub-networks have less than 100 nodes. After that, they start growing exponentially and even faster. Within the first year, all the sub-networks have more than 10000 nodes. At this size, bias in the Gini coefficient is no longer a concern. This is furthermore reflected in the evolution of the Gini coefficient of the five variables in Fig. 3. In the sub-networks, they tend to fluctuate in the first few months, but then stabilize and show actual trends. Since we are not interested in the transaction network in the first year of Blockchain, we do not have to be concerned with this instability.

---

## [Editor Report · Decision Letter 1]

16 Sep 2021

Strangely Mined Bitcoins: Empirical analysis of anomalies in the Bitcoin Blockchain Transaction Network

PONE-D-21-11382R1

Dear Dr. Óskarsdóttir,

We’re pleased to inform you that your manuscript has been judged scientifically suitable for publication and will be formally accepted for publication once it meets all outstanding technical requirements.

Kind regards,

Hocine Cherifi

Academic Editor

PLOS ONE

---

## [Editor Report · Acceptance letter]

22 Sep 2021

PONE-D-21-11382R1 

Strangely Mined Bitcoins: Empirical analysis of anomalies in the Bitcoin Blockchain Transaction Network 

Dear Dr. Óskarsdóttir:

I'm pleased to inform you that your manuscript has been deemed suitable for publication in PLOS ONE. Congratulations! Your manuscript is now with our production department. 

Kind regards, 

on behalf of

Professor Hocine Cherifi 

Academic Editor

PLOS ONE